# Coherence Characteristics of a GaAs Single Heavy-Hole Spin Qubit Using a Modified Single-Shot Latching Readout Technique

**DOI:** 10.3390/nano13050950

**Published:** 2023-03-06

**Authors:** Victor Marton, Andrew Sachrajda, Marek Korkusinski, Alex Bogan, Sergei Studenikin

**Affiliations:** Emerging Technologies Division, National Research Council of Canada, Ottawa, ON K1A 0R6, Canada

**Keywords:** spin qubit, single hole, spin readout, Rabi oscillations, Ramsey, GaAs

## Abstract

We present an experimental study of the coherence properties of a single heavy-hole spin qubit formed in one quantum dot of a gated GaAs/AlGaAs double quantum dot device. We use a modified spin-readout latching technique in which the second quantum dot serves both as an auxiliary element for a fast spin-dependent readout within a 200 ns time window and as a register for storing the spin-state information. To manipulate the single-spin qubit, we apply sequences of microwave bursts of various amplitudes and durations to make Rabi, Ramsey, Hahn-echo, and CPMG measurements. As a result of the qubit manipulation protocols combined with the latching spin readout, we determine and discuss the achieved qubit coherence times: $T_{1}$, $T_{Rabi}$, $T_{2}^{*}$, and $T_{2}^{CPMG}$ vs. microwave excitation amplitude, detuning, and additional relevant parameters.

## 1. Introduction

The GaAs material system has a long history in quantum technology [1]. The majority of the techniques currently being employed in Si and Ge electronic quantum transport devices [1] were first developed in GaAs. While for electron systems, GaAs may be considered to have been a test-bed material for hole spins, GaAs possess several favorable properties for quantum circuits and quantum platform hybridization. These include a direct bandgap for photon-to-spin conversion [2,3], strong spin-orbit coupling for fast spin manipulation and an effective hole g-factor that can be tuned in-situ using several approaches that enable addressable manipulation [4,5]. While its coherence times cannot match those of silicon or germanium, a predicted weaker interaction with nuclei would suggest an increased coherence time compared to the GaAs electron qubits [6]. With this motivation in mind, in a series of papers, we have recently studied this system employing a double quantum dot in the single and two-hole regimes [7]. To measure the critical coherence times, however, there is a non-trivial challenge. Spin-blockade techniques used frequently in electronic systems for qubit readout [8,9,10] cannot be straightforwardly adapted for holes in GaAs due to a strong tunneling spin-orbit coupling that limits the spin blockade efficiency. Additionally, the other commonly used single-shot spin readout, often called the Elzerman technique, cannot be used in situations where the spin relaxation process $T_{1}$ becomes faster than the available charge detection response time [11]. 

To solve these difficulties, we developed a new readout technique by adapting the original qubit latching technique [12]. Recently we showed [13] how it could be adapted to measure the spin relaxation time, $T_{1}$. In this paper, we demonstrate how this technique can be modified further for coherence measurements. We first measure Rabi oscillations and the corresponding coherence time $T_{R}$. Using the obtained information from the Rabi data we employ our technique to measure Ramsey oscillations and the effective coherence time $T_{2}^{*}$ of free spin precession. The coherence time $T_{2}$ was subsequently obtained by using Hahn-echo and CPMG (Carr, Purcell, Meiboom, Gill) nuclear magnetic resonance pulse sequences. These coherence characteristics of a single hole spin qubit in GaAs, have not been previously reported. Finally, we discuss the physics and the implications of the numbers obtained for $T_{Rabi}$ (~600 ns), $T_{2}^{*}$ (~15 ns) and $T_{2}^{CPMG}$ (~1 µs).

## 2. Sample and Single-Shot Readout 

Figure 1a features a scanning electron (SEM) image of the gate layout of a double quantum dot (DQD) device similar to one studied in this work but without the large global gate that lies over the imaged area that is present in the studied device. The DQD is defined in a GaAs/Al_x_Ga_1−x_As (x = 0.5) undoped heterostructure using fine surface Schottky gates fabricated by e-beam lithography. A global accumulation gate is deposited over a 110 nm Al_2_O_3_ dielectric layer and is used to generate 2D holes at the GaAs/AlGaAs interface located 65 nm below the surface [14]. Gates labeled as V_L_ and V_R_ are used to tune the energy levels of the dots and to perform the spin manipulations and readout protocols described below. Voltage detuning pulses and microwave (MW) bursts of variable amplitude durations are applied to the left control gate, V_L_. Current through the nearby charge sensor labeled in Figure 1a as I_CS_ is used to detect single-hole charges and map out the charge occupation of the DQD device as a function of V_L_ and V_R_ gate voltages in the form of a stability diagram shown in Figure 1b. The charge sensor current was amplified by a room temperature current-voltage converter (Basel Precision Instruments SP983C) and sent to a digital voltmeter set to 1 NPLC = 16 ms averaging time. More details on the device have been presented in our previous publications reviewed in ref. [7]. The device is capable of reaching the last hole regime in each dot. Figure 1b shows a stability diagram covering the four charge configurations of this regime. The vertical dash line accompanied by points is the detuning line used in the manipulation and readout protocols that will be described below. In this study, a single-hole spin qubit is formed in the left dot. This state is manipulated using MW pulses applied to the left control gate, V_L_, while the right QD is used for fast spin readout and as a memory register to store the information. 

The readout technique employed is based on the transfer of spin information into a metastable charge state via latching [13]. It takes advantage of the strong spin-orbit interaction in this system, which permits efficient non-spin conserving tunneling to take place [15]. The readout scheme principle is to transfer the higher spin state of the spin qubit formed in the left quantum dot after manipulation to the second “readout” quantum dot with a reproducible probability (~50% in our case) of transfer if the upper spin state of the qubit is occupied. During measurements, the successful transfer probability to the right quantum dot is, therefore, a direct measure of the occupancy of the upper spin state of the manipulated spin in the left quantum dot. To measure this probability, the charge state of the device is measured using the quantum point contact (QPC) charge detector via room-temperature electronics. The advantage of such a scheme [13] is that fast spin relaxations can be measured using electronics with several orders of magnitude slower time response compared to the spin relaxation process. This essentially simplifies the measurement setup removing the need for cold amplifiers or fast RF charge sensing techniques. For the experiments in this work, the modified scheme was as follows. Firstly, it was important to set the three tunneling times appropriately. The tunneling time from the left quantum dot to the left lead was set to be the fastest (*t_L_* < 20 ns). By contrast, the tunneling barrier from the right dot to the right lead was very slow (*t_R_* >> 16 ms). The tunneling time between the two dots on resonance was set to $t_{C}\approx 100\;\mathrm{ns}$. This satisfies the condition for the technique to work which is $t_{L}\ll t_{C}\ll t_{R}$. The detailed protocol used in the measurements is shown in Figure 2. The pulse sequence is illustrated in Figure 2a together with times and labels relevant to the positions in the stability diagram. In Figure 2b, the critical locations labelled in Figure 2a resulting from the pulse are indicated on the stability diagram (M—measure, T—transfer, I—initialization, and D—drive). In Figure 2c, schematics of the energy levels and the occupations during the whole sequence (stages (1) to (7)) are presented. To begin describing the sequence, we consider the occupation possibilities resulting from the previous cycle since this is required to perform the necessary initialization. At the end of any cycle (at position M), the two charge occupation possibilities are either a single hole in the right dot (0, 1) or an empty system (0, 0). For each new cycle, it is first necessary to initialize the system in the ground state with a spin-up hole in the left dot. This is accomplished by first ensuring the system is empty by transferring the hole in the right dot via the left dot to the left lead, i.e., (0, 1) to (1, 0) to (0, 0) (steps 1 through 3 in the sequence), and then adding a single hole from the left lead to the left dot. By waiting sufficiently longer than the *T*_1_ spin relaxation time, we can ensure the system is initialized in the spin-up state in the left dot (Step 4 in the sequence). In this work, we used an initialization wait time of 10 ms, which is much longer than the $T_{1}$. In Figure 2c, the schematics illustrate both readout situations i.e., where the system is in either of (0, 0) or (0, 1) at the end of the previous cycle. The single hole in the left dot is then manipulated with microwave pulses to perform specific measurements using Rabi, Ramsey, Hahn echo, or CPMG pulse sequences. This is step 5 in the diagram shown in Figure 2. At the conclusion of the microwave sequence, the readout is initiated by aligning the upper spin level in the left dot with the lower spin level in the right dot. If the upper spin level of the left dot is occupied at the end of the microwave pulses, it will have a 50% occupation probability of transferring to the right dot at the end of the alignment step (i.e., creating a (0, 1) charge occupation). If the lower spin state is occupied at the end of the manipulation, the system will remain in the (1, 0) state at the end of the transfer process. An alignment time of ~200 ns was found to be optimal (step 6 in the sequence) for the hole transfer step. At the conclusion of the transfer step, the left dot, if occupied, is emptied into the left lead (step 7 in the sequence). Thus at the end of the whole sequence there are two charge occupation possibilities (which are easily distinguished by the charge detector as described above). If the hole occupied the lower spin state (up) at the end of the manipulation, the device charge state would be (0, 0). If the upper (spin up) state were occupied, then each of the (0, 1) and (0, 0) charge configurations would be occupied with a 50% probability. The measurements involve monitoring the charge occupation of the system after each manipulation pulse. 

## 3. Rabi Oscillations

Controlled coherent rotations of a spin qubit require the application of MW bursts at the spin Larmor precession frequency $\hbar \omega_{MW}=g^{*}\mu_{B}B$, e.g., by employing hole electric dipole spin resonance (EDSR) [5]. Here $\omega_{MW}$ is the microwave cyclic frequency, $g^{*}$ is hole effective g-factor, $\mu_{B}$ is Bohr magneton, and *B* is the magnetic field normal to the sample surface. When the MW burst frequency is tuned to the hole spin Larmor frequency, spin Rabi rotations are induced between the North and the South poles of the Bloch Sphere (in the rotating frame). These rotations can be observed by measuring the spin-up probability, $P\left(↑\right),$ in the form of a 2D color map. Figure 3a shows an example of the Rabi pattern measured at B=0.896 T as a function of the MW pulse duration and frequency. The spin-up probability is an average result of 200 single-shot measurements employing the modified single-shot latching technique described above. A clear chevron-like pattern is evident in Figure 3a, as expected for Rabi oscillations. It is notable that the visibility of the oscillations quickly diminishes as the frequency is detuned away from the center line frequency *f*_0_ = 18.91 GHz.

Figure 3b presents a 2D map of the Rabi oscillations as a function of the MW pulse duration and amplitude. The amplitude is given in arbitrary units as it is technically difficult to precisely calibrate the frequency transmission of the MW lines and measure the signal amplitude reaching the control gate, V_L_. In the experiment presented in Figure 3b, the MW frequency was kept at the peak frequency while the pulse duration and the output amplitude of the MW generator were varied. As expected, the Rabi frequency increased with the MW amplitude. It is also qualitatively evident that the Rabi coherence time $T_{Rabi}$ is longer at larger MW driving amplitudes. This is consistent with Figure 3a, where the apparent coherence of the Rabi oscillations diminishes with detuning from the central frequency, which is equivalent to reducing the effective driving force.

In order to accurately determine the Rabi frequency, $f_{Rabi}$, and the Rabi relaxation time constant, $T_{Rabi}$, we conducted separate experiments close to the central frequency stepping the MW pulse duration to longer intervals, taking and averaging 500 single-shot measurements for each data point. It should be noted that the employed spin readout method requires an exact alignment of the levels at readout step 6 (Figure 2c), causing the readout fidelity to be sensitive to slow 1/f noise and drifts. To compensate for any slow 1/f noise, we repeated measurements 8 times in succession and averaged all the traces. Figure 3c presents a line graph of the Rabi frequency dependence as a function of the driving MW amplitude. In this graph, we observe a close to linear dependence of the Rabi frequency between 10 and 40 MHz. At higher MW driving amplitudes (measured in separate experiments without trace averaging, not shown) *f_Rabi_* saturated at about 50 MHz. It indicates that the hole spin π rotation can be performed in 12 ns, which is very close to the reported values of Rabi rotations in electronic GaAs DQD devices equipped with micromagnets [16]. his is an impressive result as it is expected to be more challenging to manipulate heavy hole spins due to their large g-factor anisotropy [4].

It is qualitatively evident from Figure 3b that the coherence time depends on the MW driving amplitude. It is well known that the coherence time of continuously driven oscillations differs from the $T_{2}^{*}$ and $T_{2}$ values extracted from Ramsey and CPMG experiments [1,6,17,18,19]. In order to obtain quantitative data about $T_{Rabi}$, we conducted single Rabi traces at the central frequency *f_MW_* = *f*_0_, and fitted each trace with an exponentially decaying sinusoid $P\left(↑\right)\propto \exp\left(-t/T_{Rabi}\right)\sin\left(2\pi \left(t-t_{0}\right)/w\right)$, where t is the MW pulse duration, $T_{Rabi}$ is the Rabi coherence time, $t_{0}$ is an instrumental phase factor, and w = 1/*f_Rabi_* is the Rabi period. From this procedure, we determined two fitting parameters, *T_Rabi_* and *f_Rabi_*, both of which vary with the MW amplitude. Since the MW amplitude reaching the device is less well calibrated, it is instructive to examine the dependence of $T_{Rabi}$ as a function of $f_{Rabi}$, which is plotted in Figure 3d. Note, because such a dependence of $T_{Rabi}\left(f_{Rabi}\right)$ has not yet been discussed in the literature, we approximate it with a linear dependence (dashed line). Qualitatively, we can conclude that $T_{Rabi}$ increases with the Rabi frequency, corresponding to an increasing driving amplitude. This behavior is expected if $T_{Rabi}$ is limited by the hyperfine interaction [6]. However, decoherence that occurs within a single trace is not explicitly taken into account in Ref. [6], rather, the decay is attributed to averages resulting from different hyperfine fields for each individual trace. In the data presented in Figure 3d, the linear dependence, when extrapolated to zero, intersects the Y-axis at $T_{Rabi}$ = 40 ns, not at zero. We speculate that this indicates that $T_{2}^{*}$ free evolution decoherence processes need to be taken into account to simulate the dependence $T_{Rabi}\left(f_{Rabi}\right)$ presented in Figure 3d. Further theoretical and experimental work is required to fully understand decoherence in Rabi measurements. We note that decoherence that occurs within a single trace is not explicitly taken into account in the theoretical treatment in Ref. [6]; rather, the decay is totally attributed to averages resulting from different hyperfine fields for each individual trace.

Qualitatively, it is evident in Figure 3a that the Rabi oscillations vs. time decay faster if the driving frequency is offset from the center, which is an unusual situation. To enquire quantitative data about this behavior, we conducted an additional experiment to extract data about the Rabbi coherence time vs. detuning frequency $T_{Rabi}\left(f-f_{0}\right),$ as one moves through the center frequency. In order to minimize the consequences of the non-uniform frequency transmission function of MW lines, we positioned the EDSR center frequency $f_{0}$ in the middle of the flat region of the transmission spectrum and stepped the magnetic field. Similar to the experiments presented in panels (c, d), six consecutive Rabi traces vs. MW burst duration were measured in succession and averaged for each magnetic field, which was stepped around the EDSR resonance $B_{0}=h\;f_{MW}/g^{*}\mu_{B}$, with $g^{*}$ the effective hole g-factor [20,21]. The Rabi traces were fitted with a sinusoidal decaying function as described above to extract the $T_{Rabi}$. The Rabi coherence time is plotted in Figure 3f as a function of the magnetic field. For comparison with other data, the top axis in this plot is given in frequency units using $\Delta f=(B-B_{0})g^{*}\mu_{B}/h$. It is evident from this plot that the coherence time decays sharply within a ±10 MHz range away from the center line. We speculate that this behavior is closely linked to the power dependence of $T_{Rabi}$ presented in Figure 3b,d. Detuning the MW frequency away from *f*_0_ leads to a reduced excitation strength, effectively to a reduced power. Therefore, the two phenomena presented in Figure 3d,c should be considered together to understand the underlying microscopic mechanisms affecting the Rabi coherence time.

To conclude this section, Figure 3e presents an example of a single Rabi trace with the longest coherence time $T_{Rabi}=600\pm 40\;\mathrm{ns}$ at a Rabi frequency $f_{Rabi}=38.82\pm 0.02\;\mathrm{MHz}$. It should be noted that during our experiments, we experienced occasional small telegraph noise events or small drifts requiring us to return our device to the experimental conditions required for the latching readout protocol described in Figure 2. We found that the measured Rabi coherence times were affected by these returning. Figure 3e shows an exceptionally long coherence Rabi time during which there were no drifts and no telegraphic noise events. We chose to present this result, which demonstrates that inherently $T_{Rabi}$ can be long, possibly, approaching $T_{2}^{CPMG}$ spin-echo [22] values that will be discussed in the next section. A detailed study of the decoherence factors of the Rabi oscillations is beyond the scope of this paper. As a last comment, we argue that faster readout methods and the employment of effective real-time feedback protocols can improve the measured spin coherence time [23,24].

## 4. Ramsey and CPMG Results and Discussion

Using the calibration of the π/2 and π pulses from the Rabi measurements, Ramsey experiments were conducted to determine the $T_{2}^{*}$ coherence time of the system [25,26]. Ramsey fringes experiments use two π/2 pulses separated by the wait time, $T_{wait}$. Figure 4a shows a Ramsey type measurement conducted at B=0.89 T obtained by changing the MW burst frequency of the two π/2 pulses (rotation around the X-axis of the Bloch sphere) and the delay time $T_{Wait}$ between them (rotation around the Y-axis of the Bloch sphere). Surprisingly based on our initial expectations, the coherence time of the Ramsey fringes signal is diminished for relatively small $T_{Wait}$ times ~50 ns. This measurement was repeated several times under different conditions obtained by switching between flat zones of the MW frequency transmission function, MW power, and gate detuning positions with results consistently similar to one shown in Figure 4a. The overlaid lines in blue indicate the expected fringe position following well-known Ramsey fringes formula $P\left(↑\right)\propto \cos^{2}\left(\left(f-f_{0}\right)T_{W}/2\right)$, where $f_{0}$ being the central frequency, $T_{W}$ being the wait time. This coherence time was extended by introducing one or more refocusing π pulses, corresponding to Hanh-echo and CPMG pulse sequences [27]. The Ramsey and Hahn echo experiments are presented in Figure 4a,b. Each of these plots represents the average of eleven successive Ramsey traces performed at the central frequency ($f_{MW}=18.905\;\mathrm{GHz}$) for B=0.8975 T. Figure 4b shows a Ramsey $T_{2}^{*}$ measurement at the central frequency in panel (a) stepping the wait time $T_{W}$ between the two π/2 pulses plotted on a logarithmic scale to display the large dynamic range of wait times. Each point in the plot is an average of 500 single-shot measurements of spin-up probability. The data are fitted with a decaying exponential function (solid line) which indicates a $T_{2}^{*}$ coherence time of 14 ns. We find this to be a very surprising result which is in contrast to a ten-fold $T_{2}^{*}$ enhancement over the GaAs electronic system predicted to occur due to the smaller hyperfine interaction [1,28]. We speculate that the following may contribute to the observation.

The hole quantum dot is smaller than the equivalent electron quantum dot as a consequence of the larger hole effective mass [29] $m_{HH}\approx 0.4\;m_{0}$ (electron mass in GaAs $m_{e}=0.067\;m_{0}$). Comparing images of the hole devise in Figure 1a with a typical lateral GaAs quantum dot, e.g., in Ref. [30] the lateral dimensions of the hole device are about 10 times smaller. This may be expected to lead to stronger hyperfine fluctuations experienced by the holes. The statistically fluctuating nuclear effective field is of the order of $\delta B_{N}\propto A/\left(\sqrt{N}g^{*}\mu_{B}\right)$ with A being the hyperfine interaction constant and N the number of nuclei interacting with the hole [31]. Therefore, in a hole device, since there are a smaller number of interacting nuclei, the amplitude of nuclei field fluctuations is greater. The two effects, the smaller hole hyperfine interaction and larger nuclear fluctuations in a smaller QD may be expected to compensate for one other, resulting in a similar coherence $T_{2}^{*}$ decoherence time. In other words, the advantage of holes related to their p-type orbital for hyperfine decoherence can be potentially counterbalanced by the smaller size of the hole quantum dots. An important conclusion from this observation is that larger quantum dots constructed in materials with a smaller effective mass (e.g., from strained p-type germanium [32] may be efficient in reducing hyperfine decoherence effects.

The panel in Figure 4c shows the spin-up probability for a Hahn echo experiment by introducing a π pulse between the two π/2 pulses while varying the total wait time plotted on a logarithmic scale. The data is fitted with an exponential transition function (solid line). As expected for a single π pulse the initial probability of the spin-up state is low for wait times shorter than the Hahn-echo coherence time, $T_{2}^{Hahn-echo}$. The spin-up probability increases for long wait times, eventually reaching the expected value of 0.25. It is evident from Figure 4c that the coherence time has been extended by almost two orders of magnitude to 1.15 μs by introducing just one refocusing pulse. Adding more CPMG refocusing pulses did not increase the coherence time further. The CPMG coherence time, $T_{2}^{CPMG}$, with the $N_{CPMG}=0\;\left(1\right)$ corresponding to the Ramsey (Hahn-echo) vs. $N_{CPMG}$ is presented in Figure 4d for two magnetic field magnitudes (B=0.898 T and 0.7195 T). For both data sets, the coherence time increased by approximately two orders of magnitude after the introduction of a single refocusing pulse. Such a sharp increase in the coherence time after adding just one refocusing pulse supports the argument that the short decoherence times are limited by slow hyperfine nuclei fluctuations, as discussed above.

Finally, it is important to compare $T_{2}^{*}$ and $T^{CPMG}$ coherence time constants with the longitudinal spin relaxation time $T_{1}$. We originally carried out $T_{1}$ measurements in Ref. [10], but since this was three years ago, we repeated the measurements on this particular cool down, noting that the driving/relaxation point D is much deeper in the (1, 0) region compared to our earlier measurements. The deeper position in (1, 0) could affect the spin relaxation time due to a smaller coupling to excited states, which could virtually be involved in the relaxation process. The new results of this work are compared with earlier *T*_1_ measurements in Figure 4e. The spin relaxation $T_{1}$ measurements agree closely with the previous results confirming the very good stability of the device for spin relaxation as well as the independence of $T_{1}$ on specific gate voltages (i.e., fine details of the dot potential). This is not trivial, for example, the $T_{1}$ spin relaxation time may be affected by heavy hole–light (HH-LH) hole interactions which may be expected to depend upon details of the confining potential as defined by the gate voltages. In our device, the confining potential is very shallow, resulting in the light-hole states lying above the confining DQD barrier resulting in an extended spatial distribution of the light-hole wave function. This, on one hand, minimizes the HH-LH interaction, while on the other hand, it makes certain properties of the device more predictable, including the absolute value of the effective g*-factor and it large anisotropy with respect to the magnetic field direction [4].

Theoretically, $T_{2}$ can be as long as $2T_{1}$, outlining the potential for improvement and an extension of investigations of this sample in the future. Figure 4d shows $T_{1}$ spin relaxation time measurements conducted on the same sample across various cool downs compared with the newly measured $T_{2}^{*}$ and $T_{2}^{Hahn-echo}$ coherence times. The overlaid line, drawn from previous studies of the same DQD device [13], indicates the alignment of the data measured in 2019 (shown in black) with the dominant Dresselhaus spin-orbit interaction, identified by a $B^{-5}$ dependence. The data collected via the same method in 2022 (shown in blue) exhibits consistency with these previous results despite being measured after multiple warm-up and cool-down cycles of the sample.

## 5. Conclusions

In conclusion, we have conducted the first experimental study of the coherence properties of a single heavy hole spin qubit formed in a gated GaAs/AlGaAs double quantum dot device. Holes possess a very large spin-orbit interaction, which is beneficial for fast spin manipulation techniques, but on the other hand, leads to suppression of the spin blockade phenomenon commonly used for spin-read out. We overcame this difficulty by devising a modified method to read the spin state by employing a neighboring dot in a charge-latching regime as an axillary element for fast spin readout and storing the information. The described latching method can also be adapted in other material systems, in particular, when fast spin readout protocols may be preferable [33,34]. In our work, we use this method to characterize the coherent properties of a single-hole spin qubit in GaAs via Rabi, and CPMG experiments ($N_{CPMG}=0,\;1,\;2,\;3$) with *N_CPMG_* = 0, 1 corresponding to Ramsey and Hahn echo experiments, correspondingly. Despite the p-type nature of the hole wave function, which is predicted to lead to a reduced decoherence dueto hyperfine interactions, unexpectedly, we observe very similar $T_{2}^{*}$ magnitudes to the electronic spin qubits in GaAs quantum dots. We explain this observation by the approximately 10-fold smaller size of the hole quantum dots due to larger effective mass leading to larger fluctuations $\propto 1/\sqrt{N}$ of the nuclear field bath, where *N* is number of nuclei being in contact interaction with the single hole spin. We also explored the coherence properties of a single hole spin characterizing it by various time constants, $T_{Rabi},\;T_{2}^{Ramsey}$ and $T_{2}^{CPMG}$ as a function of the MW detuning frequency, excitation amplitude, and $N_{CPMG}$. Even though the measured coherence time in our single hole GaAs device is smaller than ones reported in germanium and silicon quantum dots [25,35,36,37,38], GaAs remains a promising material for spin optoelectronic devices and applications, such as photon to spin transducers in long-range quantum repeater architectures [2,3]. The discussed microscopic physical mechanisms limiting the coherence time may also be relevant and useful for other material systems. The geometry of our dot electrostatic potential is very anisotropic [4], which is known to strongly affect coherence times, as is found in Ref. [39]. It is possible that the relaxation parameters of hole qubits in GaAs can be extended by making the QD confining potentials more circular isotropic.

## Figures and Tables

**Figure 1 nanomaterials-13-00950-f001:**
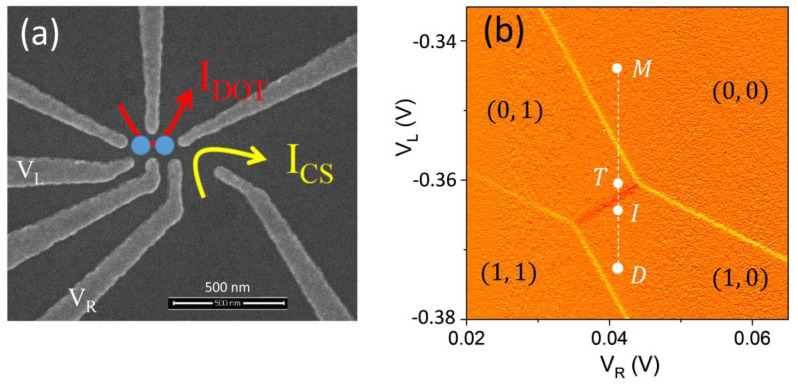
(**a**) An SEM image of the double quantum dot (DQD) device similar to the one used in this work but without the global gate laid over the fine surface gates. The yellow and red arrows show the direction of the charge sensing and inter-dot currents. The blue circles indicate the approximate locations of the two quantum dots. The labeled V_L_ and V_R_ gates are used to control the dot occupation and the detuning energy during the spin qubit manipulations. (**b**) An example of the charge stability diagram measured via the charge sensor current, dI_CS_/dV_L_. Each region is labeled with the left and right dot hole occupations (n_L_, n_R_). The detuning trajectory is indicated as the dashed white line with the detuning points of the control pulse sequence during the manipulation and readout protocol labeled as measurement (M), transfer (T), initialization (I), and (D) Drive.

**Figure 2 nanomaterials-13-00950-f002:**
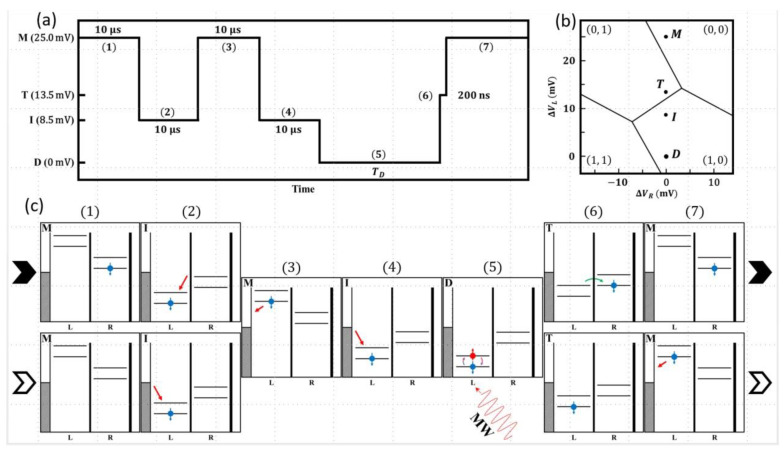
(**a**) The control pulse sequence used in the spin manipulation and read-out experiments. The steps are enumerated in brackets (N) in correspondence with the energy diagrams in (**c**). The left axis is given in millivolts of the control pulse stages labeled by a letter as measurement (M), transfer (T), initialization (I), and (D) Drive. The same letters are used in the diagrams (**b**,**c**). (**b**) A schematic of the stability diagram is shown in Figure 1b nb. The gate detuning voltages $\Delta V_{L}$ and $\Delta V_{R}$ are presented with respect to the drive point D. The vertical position of the stage points (D, I, T, M) match those marked in the Y-axis of (**a**). (**c**) The sequence of DQD energy diagrams with indicated occupations and transitions at each detuning point produced by the control pulse in (**a**). (For details see text).

**Figure 3 nanomaterials-13-00950-f003:**
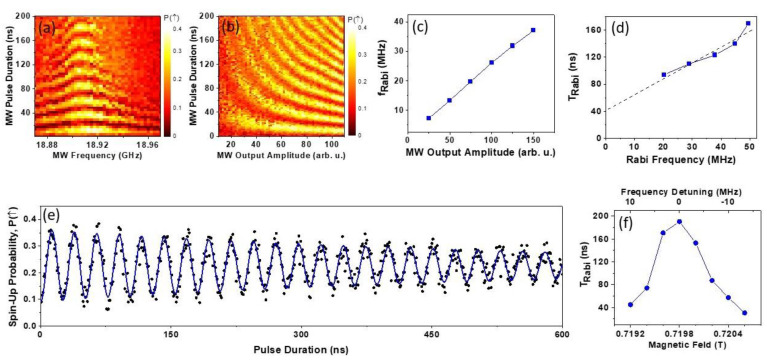
(**a**) Average spin-up probability to map out the Rabi oscillations as a function of the MW burst duration (Y-axis) and the MW frequency (X-axis); (**b**) the 2D map of the Rabi oscillations measured as the average spin-up probability in the plane of the MW Pulse Duration and MW Amplitude; (**c**) Rabi frequency as a function of MW pulse amplitude; (**d**) Period of Rabi oscillations as a function of Rabi frequency tuned by the MW burst amplitude; (**e**) An example of Rabi oscillations as a function of the MW burst duration; (**f**) Period of Rabi oscillations as a function of magnetic field stepped around the resonance frequency. The top scale shows effective frequency detuning. (For more details see text).

**Figure 4 nanomaterials-13-00950-f004:**
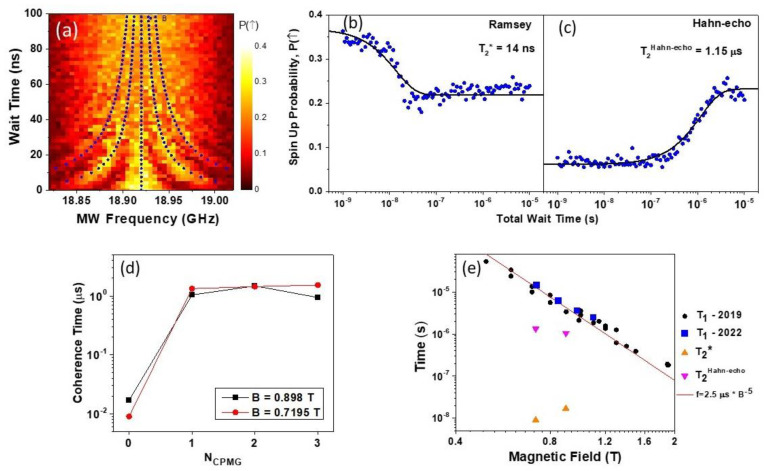
(**a**) Ramsey fringes measured by the spin-up probability using a sequence of two $\frac{\pi }{2}$ MW pulses of different frequencies (X-axis) separated by the wait time (Y-axis). Dotted lines indicate the expected position of the fringes (for more details see text); (**b**) Spin-up probability measured in the Ramsey experiment (points) fitted by an exponential decay (solid line); (**c**) spin up probability measured in Hahn echo experiment (points) fitted with an exponential step function (solid line); (**d**) coherence time for different number of refocusing CPMG pulses for two values of magnetic field (for details see text); (**e**) longitudinal spin relaxation time $T_{1}$ combined with Ramsey and Hahn echo results.

## Data Availability

The data presented in this study are available on request from the corresponding author.

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
