# Peer review of "Coherence Characteristics of a GaAs Single Heavy-Hole Spin Qubit Using a Modified Single-Shot Latching Readout Technique"

_nanomaterials, 2023, doi:10.3390/nano13050950_

Round 1

Reviewer 1 Report

The work is interesting from both a theoretical and applied point of view. However, the form of presentation is recommended to be improved (introduction..... results and conclusions)

Detailed comments:
As I pointed out, the work is of scientific interest in the field, but I believe that improvements must be made to the structure of the work (introduction-current stage-what is being pursued, materials and techniques used, results, discussions) and to the way of presentation (there are abbreviations devoted to the field, however, they must be presented more extensively in the introduction or in the other sections of the work), etc.References are recommended to be written according to the standards established by the journal.

Author Response

Response to the Referee 1 report

Thank you for your comments, which for convenience are repeated below.

The work is interesting from both a theoretical and applied point of view. However, the form of presentation is recommended to be improved (introduction..... results and conclusions)
Detailed comments:
As I pointed out, the work is of scientific interest in the field, but I believe that improvements must be made to the structure of the work (introduction-current stage-what is being pursued, materials and techniques used, results, discussions) and to the way of presentation (there are abbreviations devoted to the field, however, they must be presented more extensively in the introduction or in the other sections of the work), etc. References are recommended to be written according to the standards established by the journal.

Response:

Following your suggestion, we made a number of additions in the introduction, in the text, and in conclusions.  Following the guidelines, the changes are highlighted yellow.  We also added new references [3,11,21-23, 32-36]. 

We also introduced abbreviations when they appear for the first time, i.e.  CPMG and QPC.

We amended the references format following the NanotMaterials Template using EndNote.  We hope that the format can be easily modified to fine tune if necessary to comply with the Publisher requests.

Thank you very much once more for helpful comments.

The Authors.

Reviewer 2 Report

The draft is well-written and describes interesting research with enormous potential applications for optical and quantum technology.

Besides that, it is still room for improvement.

First, the Authors should extend the introduction and highlight the novelty of their research, not only pointing out the development of the new technique. What brings this new technique to spin-qubit physics (from fundamental understanding to potential applications)?

Second, the figure caption in Figure 2 should be enlarged and describe almost all details /information presented in this Figure.

In the end, I recommend minor but obligatory revision.

Author Response

Response to the Reviewer 2 report

The draft is well-written and describes interesting research with enormous potential applications for optical and quantum technology. Besides that, it is still room for improvement. First, the Authors should extend the introduction and highlight the novelty of their research, not only pointing out the development of the new technique. What brings this new technique to spin-qubit physics (from fundamental understanding to potential applications)?
Second, the figure caption in Figure 2 should be enlarged and describe almost all details /information presented in this Figure.  In the end, I recommend minor but obligatory revision.

Response:

Thank you for your constructive comments.
Following your recommendations, we have introduced additional explanations highlighting the novelty of this work.   (For convenience the additional text is highlighted yellow). We also added more references to relevant works required in the extended material [3,11,21-23, 32-36]. 

In regards of Figure 2, thank you for pointing to this omission.  The caption has been appropriately expended to describe all details shown in the Figure. 

The Authors.

Reviewer 3 Report

The authors have conducted an experimental study of the coherence properties of a single heavy hole spin qubit formed in a gated GaAs/AlGaAs double quantum dot device. It sounds technical and well written, so I recommend accepting it after a few revisions.

1. Reference numbers, section numbers, and figure boxes should be modified according to the format of the journal.

2. Fig. and figure are mixed and used, so it is recommended to unify it as a figure.

3. The pros and cons and limitations of existing related studies should be explained, and how the proposed study overcomes the limitations should be clearly shown.

4. Many typing mistakes should be corrected through thorough reading (ex. Indentation of the second line of INTRODUCTION, Spacing, Sentence punctuation CONCLUSSION, etc.)

Author Response

Response to the Reviewer 3 report.

The authors have conducted an experimental study of the coherence properties of a single heavy hole spin qubit formed in a gated GaAs/AlGaAs double quantum dot device. It sounds technical and well written, so I recommend accepting it after a few revisions.

  1. Reference numbers, section numbers, and figure boxes should be modified according to the format of the journal.
    2. Fig. and figure are mixed and used, so it is recommended to unify it as a figure.
    3. The pros and cons and limitations of existing related studies should be explained, and how the proposed study overcomes the limitations should be clearly shown.
    4. Many typing mistakes should be corrected through thorough reading (ex. Indentation of the second line of INTRODUCTION, Spacing, Sentence punctuation CONCLUSSION, etc.)

Response:

Thank you for your helpful comments and concrete recommendation.

  1. We have modified Reference numbers, section numbers and figure boxes following the NanoMaterials Template. We also have rebuilt the Reference list using EndNote to follow the NanoMaterials recommended style.
  2. We have adopted the same style along the text to reference Figures.
  3. We have introduced additional explanations to address this comment. The added text is highlighted yellow in the new version of the manuscript.
  4. We have carefully proofread the manuscript and corrected all typing and formatting errors that we could spot.

After these corrections, we hope that the paper is suitable for the publication.

The Authors

Round 2

Reviewer 1 Report

see the attachment
